# Using 0.1 THz Radiation Regulates Collagen Synthesis Through TGF-β/Smad Signaling Pathway in Human Fetal Scleral Fibroblasts

**DOI:** 10.3390/cells14191512

**Published:** 2025-09-28

**Authors:** Wenxia Wang, Liu Sun, Lei Wang, Jinwu Zhao, Shuocheng She, Pandeng Hou, Mingxia He

**Affiliations:** 1The Center for Terahertz Waves, School of Precision Instrument and Opto-Electronics Engineering, Tianjin University, Tianjin 300072, China; wwx_0817@163.com (W.W.); 13103249606@163.com (L.S.); wonglei@tju.edu.cn (L.W.); zhaojinwu@tju.edu.cn (J.Z.); syc951011@tju.edu.cn (S.S.); howa2d@tju.edu.cn (P.H.); 2State Key Laboratory of Precision Measuring Technology and Instruments, Tianjin University, Tianjin 300072, China

**Keywords:** human fetal scleral fibroblasts, terahertz radiation, extracellular matrix remodeling, proteomics analysis, bioinformatics analysis

## Abstract

Scleral tissue is a connective tissue made up of dense, intertwined collagen fibers that plays a vital part in preserving both the integrity of vision and the shape of the eyeball. Numerous studies have been conducted on the impact of terahertz radiation on biological systems. Terahertz radiation can affect cell morphology and function by mediating modifications in protein conformation and gene expression, according to recent research. Though terahertz waves found in the environment directly expose scleral tissue, little is known about how terahertz radiation affects scleral fibroblasts biologically. In this work, we investigated how 0.1 THz radiation affected the global expression levels of proteins and the viability of human fetal scleral fibroblasts (HFSFs). A total of 79.44% of the differentially expressed proteins (DEPs) showed significant downregulation in expression levels after 60 min of exposure to terahertz radiation. Enrichment analysis of DEPs revealed that terahertz radiation enhanced the expression of cytoskeletal keratins, disrupted supercoplexes’ assembly, and impaired mitochondrial respiration. Moreover, terahertz radiation influences the remodeling process of the scleral extracellular matrix by triggering the TGF-β/Smad signaling pathway. Changes in transcriptional activity of several extracellular matrix (ECM)-related genes persisted for 12 h in the absence of terahertz radiation. Research findings indicate that 0.1 THz radiation is capable of disrupting the dynamic balance between collagen synthesis and degradation in scleral fibroblasts. Such an imbalance may induce alterations in the structural integrity and biomechanical properties of the sclera, thereby elevating the potential risk of myopia onset or progression.

## 1. Introduction

Terahertz wave is an electromagnetic wave that has a frequency between 0.1 and 10 THz. The low photon energy of these waves makes them insufficient to ionize biological things [1,2,3]. Researchers are becoming more interested in examining the connection between terahertz radiation and biological effects because of the widespread application of terahertz radiation sources in industries including communications, security, and medicine. According to several published studies, organisms are not significantly harmed by terahertz radiation. Amici and Franchini et al. exposed human fetal and adult fibroblasts to broadband terahertz radiation (0.1–0.15 THz, 0.40 mW/cm^2^) for 20 min, an infrared camera showed no temperature increase of more than 0.2 °C. This temperature change was not enough to cause a noticeable rise in the expression of the heat shock proteins (HSPs), and both results show that no DNA damage was caused. Nonetheless, some research has come to the opposite result [4,5]. Wilmink et al. found that heat shock proteins (HSPs) were expressed more when they exposed human skin fibroblasts (HDF cell line) to continuous terahertz radiation (frequency 2.52 THz, power density 85 mW/cm^2^). This was primarily brought on by the cell culture’s temperature rising to 40 °C (it was 37 °C at the beginning of the experiment and then increased by 3 °C during radiation due to the effects of terahertz radiation) while exposed to terahertz radiation, but the outcome was identical to that with conventional heating, and additional detection results demonstrated that the radiation conditions had no appreciable impact on the cellular and molecular levels [6]. Although the precise mechanisms of these reactions are yet unknown, the majority of recent study reports suggest that terahertz radiation can affect biological systems in a variety of ways through non-thermal effects.

Since skin tissue absorbs the majority of terahertz radiation, skin fibroblasts are utilized as cell models in the majority of contemporary studies [7]. The connection between terahertz radiation and the skin has been verified by a number of studies. Terahertz radiation has an impact on how the skin functions, causing inflammation, controlling the immune system, and accelerating the healing of wounds. Kim et al. found that fs-THz radiation (2.5 THz, 0.32 μW/cm^2^) interacted with the wound healing process by activating the TGF-β signaling pathway and altering gene expression, which interfered with the wound healing process [8]. According to Lin et al., mice exposed to terahertz radiation (2.5 THz, 100 mW/cm^2^) saw faster wound healing and higher collagen III levels. The predominant cell type in scleral tissue is scleral fibroblasts [9]. TGF-β is one of the efficient regulators of scleral extracellular matrix (ECM) remodeling, contributing mainly to the remodeling process of scleral ECM through collagen synthesis and degradation, according to several studies on TGF-β and myopia [10,11]. Since the scleral tissue is directly exposed to terahertz radiation from the surroundings, it is an essential subject for research on how terahertz radiation affects eye health.

In this work, the expression levels of global proteins in HFSFs exposed to terahertz radiation (0.1 THz, 44 mW/cm^2^) were determined using proteomics analysis. We performed functional annotation and enrichment analysis of the proteins that were differently expressed in the control group and the terahertz radiation group using bioinformatics analytic techniques. Terahertz radiation time and the expression of genes associated with the extracellular matrix of HFSFs were determined to be connected using real-time fluorescent quantitative PCR (qPCR) technology. The current investigation may provide new understandings of the molecular processes by which terahertz radiation controls cell activity. This study also suggests recommendations for controlling the progression of myopia by the use of terahertz radiation.

## 2. Materials and Methods

### 2.1. Cell Culture

Human fetal scleral fibroblasts (HFSFs) were purchased from the Cell Resource Center, Peking Union Medical College (which is part of the National Science and Technology Infrastructure, the National Biomedical Cell-Line Resource, NSTI-BMCR. http://cellresource.cn). The HFSFs were cultured in Dulbecco’s modified Eagle’s medium (DMEM, Corning Inc., Corning, NY, USA) supplemented with 10% Fetal bovine serum (FBS, Christchurch, New Zealand) and 1% penicillin-streptomycin solution (Gibco, Waltham, MA, USA) and 2 mM L-Glutamine Liquid (Beijing Solarbio, Science & Technology Co., Ltd., Beijing, China) in culture dishes. The culture medium was changed every 48 h until the culture reached 80% confluence, and the cells were maintained at 37 °C and 5% CO_2_ in a cell incubator (Thermo Fischer Scientific, Waltham, MA, USA). Trypsin (Gibco, Inc., Corning, NY, USA) digestion was performed on HFSF cells, which were transferred to polystyrene 24-well plates (Corning Inc., Corning, NY, USA) 12 h prior to the terahertz radiation exposure experiment. Each well received 1 mL of culture medium, and the cells were introduced at a density of 2 × 10^5^ cells/well.

### 2.2. THz Sources and Irradiation

Figure 1 displays the schematic diagram of the complete terahertz radiation system used in this investigation. The TeraSense series of terahertz sources (IMPATT diodes) was used to expose HFSFs at a frequency of 0.1 THz and a power of 88 mw. The HFSF-inoculated 24-cell culture plate (Corning Inc., Corning, NY, USA) well plates were put in a terahertz radiation chamber that was 37 °C inside. Polystyrene has been demonstrated to have minimal material loss (with a refractive index ranging from 1.017 to 1.022 and a dielectric constant of approximately 2.45–2.65), thus rendering it an optimal material for terahertz (THz) applications. [12,13]. The terahertz radiation optical path system is composed of two off-axis parabolic mirrors (OPA) and a reflector. The utilisation of Golay detectors (TYDEX, GC-1P) in conjunction with a metal diaphragm facilitated the determination of a 1.8 cm-diameter spot position at the terminus of the terahertz beam. The distance from the output port of the terahertz source to the cell culture dish was measured using optical geometry methods, yielding a value of approximately 18.07 inches. The terahertz beam was then directed by a reflector to propagate vertically upward, ultimately irradiating the HFSFs located at the bottom of the 24-well culture plate. The endpoint diameter of the terahertz wave was measured with geometric optics and found to be 1.6 cm with an average power density of 44 mW/cm^2^. To prevent significant attenuation of the terahertz waves during propagation, the humidity of the environment where the terahertz optical route system is located is maintained below 5% during the experiment. We employed a four-terminal thermocouple thermometer (SNART SENSOR, AS887, Accuracy: ±1.5%) to monitor the temperature changes in the 0.1 THz radiation culture medium over 120 min. In the wells designated for the radiation and control of the 24-well cell culture plate, 1 mL of complete culture medium should be added. The terminal extremity of the thermocouple probe is then to be positioned at the nadir of the well. The results are shown in Figure 2B.

### 2.3. Cell Viability Assay

HFSF cell viability was determined by using the Cell Counting Kit-8 (CCK-8, meilunbio, MA0218). HFSF cells were inoculated at a density of 8 × 10^4^ cells/well in 48-well cell culture plates (Corning Inc., Corning, NY, USA), with two wells separating the radiation and control groups. After 60, 120, and 180 min of exposure to terahertz radiation, each culture well received 20 µL of CCK-8 solution, which was then cultured for 120 min in an incubator. An EnSpire Multimode Plate Reader (PerkinElmer, Waltham, MA, USA) was used to detect absorbance at 450 nm. For the validation of each radiation duration, 3 independent biological samples were used; for each independent sample, the absorbance value at OD450 was measured in triplicate using a microplate reader to reduce instrument measurement error. Figure 2A shows the change in HFSF cell viability over time with terahertz radiation.

### 2.4. Proteomics Analysis

After the completion of the terahertz radiation research, HFSFs were digested using trypsin (Gibco, Waltham, MA, USA), and the precipitated cells were then collected. The proteome was analyzed by Scale Biomedicine Technology Co., Ltd. (Beijing, China). HFSFs exposed to terahertz radiation are lysed using a denaturing buffer that comprises 100 mmol/L of triethylammonium carbonate (Thermo Fischer Scientific, Waltham, MA, USA, T7408-500 ML) and 8 moles/liter of urea (Sinopharm Chemical Reagent, 10023218). The Bradford protein quantitative kit (Beyotime, Songjiang, Shanghai, China) protocol was followed in order to quantify the total protein concentration.

After trypsin-gold (Promega, Madison, WI, USA, V5280) was used to digest these protein samples, they were slowly put onto a C18 deionization column to desalt them. Every sample that eluted was collected and lyophilized. Using a Q Exactive^TM^ series mass spectrometer and an EASY-nLC^TM^ 1200 UHPLC system (Thermo Fisher Scientific, Waltham, MA, USA), three proteomic analyses were carried out on each sample. Spectronaut 17 software (Biognosys AG, Schlieren, Switzerland) was used to process and analyze the raw data of liquid chromatography tandem mass spectrometry (LC-MS/MS) detection using the default parameters. Download the Uniprot/NCBI database’s Homo sapiens protein sequence database (FASTA). Trypsin/p was set as the digestion enzyme specificity, and specific was set as the digest type.

The search criteria included carbamidomethylation of cysteine as a fixed modification and oxidation of methionine as a variable modification. Dynamic iRT (the normalized retention time) was selected as the retention time prediction type. Depending on gradient stability and iRT calibration, the ideal data extraction window is dynamically determined. At the protein and peptide precursor levels, the Q-value cutoff was set at 1%. The decoy generation was set to mutated, which applies a random number of AA position swamps (min = 2, max = length/2) and is similar to scrambled. Local normalization was used as the normalizing strategy. The MaxLFQ method was used to calculate the major group quantities for peptides that passed the 1% Q-value cutoff.

The function and characteristics of proteins in HFSF cells were determined using bioinformatics analysis. Gene Ontology (GO) and Kyoto Encyclopedia of Genes and Genomes (KEGG) used DAVID (https://davidbioinformatics.nih.gov/, access Date: 2 June 2025) to assess differentially expressed proteins. Using the Search Tool for Interacting Genes (STRING) (https://cn.string-db.org/, access Date: 21 March 2025), a protein–protein interaction (PPI) network was constructed using 214 differentially expressed proteins (FC ≥ 1.6, FC ≤ 0.625, *p* < 0.05) from scleral fibroblasts. The minimum required interaction score was set at 0.900 (the highest confidence level), and disconnected nodes were hidden inside the network. The software Cytoscape (version 3.7.0) was used to calculate the nodes and visualize the constructed network. The betweenness centrality (BC) of the nodes is calculated to determine the size of the protein node.

### 2.5. RNA Extraction and qPCR

Following the manufacturer’s instructions, total RNA was isolated from HFSFs using Total RNA Extraction Reagent (TRIzol, ABclonal, Wuhan, Hubei, China, RK30129). The concentration of total RNA was measured using a NanoDrop One Microvolume UV-Vis Spectrophotometer (Thermo Fisher Scientific). The GoScript^TM^ Reverse Transcription System (Promega, A5000) was used to reverse-transcribe the RNA samples. The qPCR analysis was performed using the GoTaq qPCR Master Mix (Promega, A6001) and LightCycler^®^96 (Roche Diagnostics Applied Science, Basel, Switzerland). The qPCR reaction parameters were set as follows: initial denaturation: initial denaturation at 95 °C for 10 min, followed by 40 cycles of denaturation at 95 °C for 15 s, and annealing/elongation at 60 °C for 60 s. Table 1 lists the primer sequences for the gene-specific primers that were designed and procured from Takara Bio (Dalian, China). The 2^−ΔΔCt^ method was used to calculate the relative expression levels of the target genes. Glyceraldehyde-3-phosphate dehydrogenase (GAPDH) was selected as the internal control, and at least three biological replicates of each experiment were conducted. For each radiation exposure duration, three independently cultured biological samples were utilized. For each target gene, detection was performed in three independent wells (serving as technical replicates), and the mean value was computed to serve as the analytical data for both the experimental group and the control group.

### 2.6. Statistical Data Analysis

This study employed GraphPad Prism 9.50 software for data analysis and statistical evaluation. Initially, Levene’s test was performed on data from the radiation group versus the control group, as well as among groups with different radiation durations, to verify the assumption of variance homogeneity. If the Levene test yields a *p*-value ≥ 0.05, indicating that the data satisfy the assumption of variance homogeneity, the independent samples *t*-test (Student’s *t*-test) is employed for intergroup comparisons. Conversely, if the Levene test yields a *p*-value < 0.05, indicating that the assumption of variance homogeneity is violated, the Welch *t*-test is utilized for analysis. Both Student’s *t*-test and Welch’s *t*-test are predicated on the assumption that potential differences exist between the two groups of data. The criteria for interpreting statistical results are as follows: if either Student’s *t*-test or Welch’s *t*-test yields a *p*-value ≥ 0.05, the intergroup difference is considered statistically non-significant, and such results are not presented in this study; if *p* < 0.05, the intergroup difference is deemed statistically significant. Data are expressed as mean ± standard deviation. Statistical significance in the text is denoted using the following conventions: * denotes *p* < 0.05, ** denotes *p* < 0.01, *** denotes *p* < 0.001, and **** denotes *p* < 0.0001.

## 3. Results

### 3.1. The Vitality of HFSFs Is Improved by Terahertz Radiation

The vitality of HFSFs was altered by terahertz radiation for 60, 120, and 180 min, but no cytotoxicity was observed when compared to control cells that were not exposed to the radiation (Figure 2A). The viability capacity of HFSFs increased by 14.98 ± 4.16 after 60 min of terahertz radiation exposure. Long-term terahertz radiation exposure, however, lessened this effect. Its vitality peaked at 103.52 ± 9.70 after 120 min, and then it recovered to normal after 180 min.

### 3.2. The Different Expression of Proteins Induced by THz Irradiation

Global proteins in the control group and the THz radiation group had their expression levels analyzed using proteomics techniques in order to investigate the effects of 0.1 THz radiation on HFSFs. The fold change (FC) was computed as the ratio of the mean values of the two sample groups, and the *p*-value was computed as the significance of the differences in protein expression between the two sample groups. Using *p* < 0.05 as the selection criterion for valid protein data, 3055 proteins in total were screened. A total of 214 proteins were found using FC ≥ 1.6 and FC ≤ 0.625 as the selection criteria for DEPs. Of these, 44 were up-regulated (making up 20.56% of DEPs) and 170 were down-regulated (making up 79.44% of DEPs) (Figure 2B). Owing to the lack of multiple testing correction, the list of differentially expressed proteins identified here is subject to false positives. The number of downregulated proteins was 3.86 times more than the number of upregulated proteins, according to our findings. This finding suggests that 0.1 THz radiation may influence the expression levels of multiple proteins in HFSFs, which could potentially impact cellular function. Nevertheless, additional experimental verification is necessary to delineate the association between these differentially expressed proteins and terahertz radiation; this step is critical for elucidating the underpinning mechanisms and validating their physiological relevance.

### 3.3. Heat Shock Proteins Do Not Respond to Terahertz Radiation

During the irradiation process, the temperature of the experimental group was recorded as approximately 34.38 ± 0.80 °C, whereas that of the control group was measured to be around 33.07 ± 0.31 °C. The mean temperature difference between the two groups was approximately 1.32 ± 0.64 °C. Throughout the entire irradiation period, the temperature of the culture medium remained below 37 °C, a phenomenon that can be attributed to its exposure to the ambient environment. At the commencement of the measurement process, the experimental group exhibited a lower temperature in comparison with the control group. This observation can be attributed to the placement of the experimental group directly beneath the terahertz radiation aperture, which thereby precluded contact with the heating apparatus.

Using *p* < 0.05 as the screening criterion, proteomics analysis found 3055 proteins in total. These were compared with 109 heat shock protein data (Relevance score ≥ 18) retrieved from the GeneCards database (http://www.genecards.org, access Date: 1 April 2025) (Appendix A). After screening for 71 heat shock-related proteins, only 8 were discovered to have elevated protein levels (Figure 2C). A fold change of 1.40 was seen in the level of the 70 kDa heat shock protein 6 (HSPA6). The results indicate that 0.1 THz radiation does not induce significant thermal effects during the process, and consequently does not cause pronounced severe apoptosis in cells.

### 3.4. Gene Ontology (GO) Analysis

To determine the functionalities of the identified DEPs (FC ≥ 1.6 FC ≤ 0.625 and *p* < 0.05), gene ontology (GO) enrichment analysis was performed with the DAVID database. These functions comprised molecular function, biological process, and cellular component. Ten annotation clusters were identified by enrichment analysis of the functions of DEPs using the DAVID database’s “Functional Annotation Clustering” module (Appendix A). These clusters of annotations were combined according to the cellular component. According to the results, the cytoskeleton, mitochondria, ribosomes, Golgi bodies, and chromosomes were the primary sites of DEPs caused by 0.1 THz radiation (Figure 3A–E). These DEPs are mostly involved in biological processes such as mitochondrial adenosine triphosphate (ATP) synthesis, structural components of ribosome, protein transport, structural components of cytoskeletal, and chromatin remodeling.

### 3.5. KEGG and Disease Annotation Analysis

A total of 5 Disease Annotation Clusters and 2 KEGG Annotation Clusters were produced by cluster analysis of the DEPs using the DAVID database’s “Functional Annotation Clustering” module (Figure 3F,G) (Appendix A). KEGG cluster enrichment analysis identified several DEPs that are associated with the oxidative phosphorylation and mitochondrial metabolic pathways. These DEPs may be associated with neurological disorders and various infectious diseases that are related to endoplasmic reticulum protein processing. Additionally, in the disease annotation clustering analysis, certain protein features were observed to be associated with Pachyonychia Congenita, Leigh syndrome, Malignant neo-plasm of stomach, Hepatitis and Breast Carcinoma. Notably, terahertz radiation may be associated with the overexpression of KRT16, KRT6B, and KRT6A proteins, which can serve as primary candidate proteins for studies investigating the relationship between terahertz radiation and two conditions: palmoplantar keratoderma and Pachyonychia Congenita. However, these preliminary proteomic findings require validation in more extensive physiological systems or disease models to evaluate their potential as biomarkers for terahertz bioeffects.

### 3.6. Protein–Protein Interaction (PPI) Analysis

A network comprising 53 nodes was formed from 214 DEPs (FC > 1.6, FC ≤ 0.625, *p* < 0.05) that were selected for protein–protein interaction (PPI) research (Figure 4 and Appendix A). These nodes’ proteins are mostly found in the mitochondria, ribosomes, and cytoskeleton. Only 5 of these node proteins represent enhanced expression, including Keratin16 (KRT16, FC = 1.77), Keratin6B (KRT6B, FC = 2.04), Keratin6A (KRT6A, FC = 2.23), Large ribosomal subunit protein bL34m (MRPL34, FC = 2.04), and Ubiquitin-like FUBI-ribosomal protein eS30 fusion protein (FAU, FC = 1.73).

These 3 protein clusters are mostly involved in organelles such the cytoskeleton, ribosomes, and mitochondria. High levels of protein expression are found in cytoskeletal proteins, such as KRT16 and its companion proteins KRT6A and KRT6B. Because of the very small interaction networks formed by these three proteins, KTR16 has strong betweenness centrality. The PPI Network’s ribosomal protein cluster is made up of two components: ribosomal protein and mitochondrial ribosomal protein. These DEPs are involved in a variety of biological processes, including proton transmembrane transport, proton potential-driven ATP synthesis, aerobic respiration, and mitochondrial respiratory chain complex I assembly. The PPI Network’s mitochondrial protein cluster involves complex I (mitochondrial NADH dehydrogenase subunit 3 [MT-ND3] and MT-ND6), complex IV (mitochondrial cytochrome oxidase subunit 2 [MT-CO2] and MT-CO3), and complex V (mitochondrial ATP synthase subunit 6 [MT-ATP6] and MT-ATP8). The remaining proteins are post-translationally transmitted to the mitochondria to support mitochondrial survival after being encoded by nuclear DNA (nDNA) [14]. The study’s findings suggest that 0.1 terahertz radiation may interfere with the formation of mitochondrial respiratory chain complexes, which could lower ATP synthesis via oxidative phosphorylation and impact HFSF metabolism and other biological processes.

### 3.7. Terahertz Radiation Activates the Transforming Growth Factor-β/Smad Signaling Pathway

The DAVID database’s GO enrichment analysis was used to further investigate the effect of 0.1 terahertz radiation on scleral fibroblast functions by enriching the 1883 DEPs (FC ≥ 1.2, FC ≤ 0.83, *p* < 0.05). 30 positive regulation terms and 11 negative regulation terms were among the 344 biological processes that were enriched in total. The top 7 biological processes with positive and negative regulation terms are displayed in Figure 5A, respectively. Analyzing these connected terms helps us better understand how terahertz radiation affects the regulation of biological processes. Interestingly, the apoptotic process positive regulation and negative regulation is enriched in GO enrichment analysis.

The results of this research suggest that 0.1 THz radiation can activate HFSFs’ apoptotic process, while cellular immune regulating mechanisms may control this effect. Additionally, the TGF-β/Smad signaling pathway is important for extramural scleral matrix remodeling, and Smad protein signal transduction is a key mechanism for moving TGF-β superfamily signals from the cell surface to the cell nucleus. The Smad signaling pathway enriched nine proteins in total, including two up-regulated proteins (Disabled homolog 2 [DAB2, FC = 1.22] and SH2B adapter protein 1 [SH2B1, FC = 1.53]) and seven down-regulated proteins (TGF-beta receptor type-1 [TGFBR1, FC = 0.69], TGF-beta receptor type-2 [TGFBR2, FC = 0.77], Mothers against decapentaplegic homolog 3 [SMAD3, FC = 0.72], Nuclear pore complex protein Nup93 [NUP93, FC = 0.82], D-glucuronyl C5-epimerase [GLCE, FC = 0.74], Casein kinase II subunit beta [CSNK2B, FC = 0.76] and Tyrosine-protein kinase JAK2 [JAK2, FC = 0.77]). According to these findings, the TGF-β/Smad signaling pathway may be positively regulated by 0.1 THz radiation, which might influence the extramural scleral matrix remodeling process.

### 3.8. Terahertz Radiation Inhibits Collagen Synthesis and Degradation in HFSFs

The present study investigates how terahertz radiation regulates the expression of genes (COL1A, MMP-2, TIMP-2, TGF-β2) associated with the extracellular matrix of HFSFs. HFSFs were exposed to 0.1 THz radiation for 30, 60, and 120 min, respectively. To explore the effect of radiation duration on changes in gene expression in HFSFs, qPCR analysis was performed immediately following exposure (Figure 5C–F). The 12-st group meant of the HFSFs that were exposed to 0.1 THz radiation for 120 min and subsequently cultivated for 12 h in a cell culture incubator. This group was created to assess the long-term effects of 0.1 THz radiation on HFSF gene expression. Following 60 min of terahertz radiation, the qPCR detection results demonstrated that the protein and gene expression levels of COL1A1, MMP-2, TIMP-2, and TGF-β2 in HFSFs remained basically consistent. This indicates that the proteomics analysis results in this study were reliability.

In HFSFs, COL1A1 gene expression decreased to 0.93 after 120 min of terahertz radiation exposure (*p* < 0.05). However, in the absence of terahertz radiation, this effect subsided within 12 h. According to proteomics analysis, 60 min of terahertz radiation raised the level of expression of the Transforming Growth Factor Beta-2 proprotein (TGFB2, FC = 1.18) (Figure 5B); nevertheless, TGF-β2 gene expression (relative expression = 0.93), on the other hand, slightly decreased. In these conditions, the relative gene expression level of TGF-β2 increased after 120 min of radiation (relative expression = 1.06), and this impact lasted for 12 h. According to the study, MMP-2 and TIMP-2 genes were somewhat upregulated in HFSF cells exposed to terahertz radiation for 120 min, although this effect was not significant. The 120-st group displayed an intriguing pattern: MMP-2 expression levels slightly increased while TIMP-2 expression levels returned to normal. According to these findings, 0.1 THz radiation could regulate gene expression in HFSFs, hence influencing to the extra-scleral matrix remodeling process.

## 4. Discussion

This proteomics investigation uses proteomics and bioinformatics approaches to analyze the total protein expression status of HFSHs exposed to 0.1 THz radiation. It offers fresh perspectives on how terahertz radiation affects cell function. There were no negative effects of terahertz radiation on cell viability or high expression of heat shock proteins in this investigation. These findings are in line with a number of recent investigations that show non-thermal effects are the main cause of terahertz radiation-induced changes in protein expression [15,16]. In short, the radiation parameters used in this investigation are not thought to be thermal stimulation for HFSFs. 79.44% of the DEPs (FC ≥ 1.6, FC ≤ 0.625, *p* < 0.05) showed significant downregulation in expression levels after 60 min of exposure to 0.1 THz radiation in HFSFs. These proteins can be found in subcellular compartments such the cytoskeleton, mitochondria, ribosomes, the Golgi apparatus, and chromosomes. Through their interactions, these differently expressed proteins within the cell create a complex network that regulates a variety of biological functions. Studying the relationship between these proteins with different expressions and cellular processes is essential to identifying the molecular mechanism of terahertz radiation’s cellular effects.

The type I intermediate filament cytoskeletal protein KRT16 is crucial for maintaining the integrity and structural stability of cells [17,18]. According to results of the GO analysis, KRT16, KRT6B, and KRT6A protein are connected to biological processes including adaptive immune responses and keratinization. Healthy human eyes contain very little keratin, however noninfectious scleritis and myopic eyes have been found to have several keratin family members [19,20]. The role and mechanism of these keratins in scleral tissue are still unknown, nevertheless. In skin tissue, KRT6 and its polymerization partner KRT16 are considered to be damage-associated keratins, and wounds cause a considerable increase in their expression [21,22]. By controlling cell migration and substrate adherence, these keratins contribute significantly to tissue regeneration. The connection between trauma response and terahertz radiation has been documented in earlier research. Kim et al. found that mouse skin punch wounds were delayed in closing after being exposed to repeated femtosecond-terahertz (fs-THz, 2.5 THz, 0.32 μW/cm^2^) pulse radiation8. According to Lin et al. terahertz radiation (2.52 THz, 110 mW/cm^2^) considerably accelerated the rate at which mice’s acute wounds healed9. There is no direct evidence that 0.1 THz radiation has an impact on wound healing, according to this study. The markedly elevated keratin expression levels, however, provide a fresh viewpoint on the connection between terahertz radiation and wound healing.

Protein synthesis takes place in ribosomes, and ribosomal biogenesis is crucial for orchestrating the functioning of major cellular processes [23]. According to the GO analysis’s findings, these ribosomal subunit proteins are closely associated with a number of biological processes, such as RNA binding, cytoplasmic translation, and mitochondrial translation. Ribosomal protein levels were significantly reduced in HFSFs exposed to 0.1 THz radiation, indicating that terahertz radiation could affect cellular biological processes by affecting ribosome protein production. According to Bogomazova et al. terahertz radiation (2.3 THz, 0.14 W/cm^2^) increased the expression of several genes encoding mitochondrial ribosomal components in human embryonic stem cells [24]. Shang et al. reported that terahertz radiation (0.1 THz, 33 mW/cm^2^) significantly upregulated and enriched the genes for mitochondrial and ribosomal proteins in primitive hippocampal neurons [25]. These results of the present research indicate that ribosome gene and protein expression are sensitive to terahertz radiation. Understanding how terahertz radiation regulates ribosome biosynthesis is essential to revealing the way it affects biological processes.

A total of 0.1 THz radiation caused the expression of several proteins involved in mitochondrial function in HFSFs. These mitochondrial proteins are the subunits of complexes I, IV and V. The inner membrane of the mitochondria contains the OXPHOS system [14]. The function of this system is responsible for generating ATP by the coordinated transfer of electrons through complexes I–V. Lei et al. found that human neuroblastoma-like cells (SH-SY5Y) exposed to 0.1 THz radiation showed increased activity of complexes I and V but no changes in activity of complexes II, III, and IV. The report’s other research verified that terahertz radiation impairs mitochondrial respiration and disrupts super complex assembly structures [26]. Their findings were consistent with our research. This is due to the work’s discovery that terahertz radiation inhibits the expression of mitochondrial proteins that are essential for biological functions such proton transmembrane transport, proton motive force-driven ATP synthesis, aerobic respiration and mitochondrial respiratory chain complex I assembly. Furthermore, these mitochondrial proteins are enriched in neurodegenerative signaling pathways-multiple diseases, including Parkinson’s, Alzheimer’s, Huntington’s, and amyotrophic lateral sclerosis [14,20]. Research on scleral tissue has been lacking, despite earlier findings that neurodegenerative diseases result in the degeneration of specific retinal neurons [27]. It is noteworthy that specific terahertz radiation may regulate Alzheimer’s disease [28,29]. Consequently, our work indicates that terahertz radiation could affect mitochondrial protein expression levels, thereby regulating neurodegenerative diseases. Additionally, pathways of neurodegeneration were enriched in the proteomic data of the vitreous and retina of animal models of myopia [30,31]. This finding suggests the potential for inducing myopia formation under the radiation conditions employed in this study. However, further research is necessary to substantiate this hypothesis.

Transforming growth factor-β, or TGF-β, has a wide range of biological functions and actions in various types of cells and is essential to regulate immunological response, extracellular matrix deposition and cell growth [32]. According to several results in animal models, the TGF-β signaling pathway is a crucial signaling pathway that is triggered by various terahertz radiations. These models include mouse skin [8], arthritic mice [33], and Caenorhabditis elegans [15]. The results of this study indicated that Smad protein signal transduction was activated when DEPs (FC ≥ 1.2, FC ≤ 0.83, *p* < 0.05) were annotated using gene ontology (GO) enrichment analysis. The Smad proteins are a family of transcription factors that have been identified as the only receptor substrates with the ability to transmit TGF-β signaling [34]. The activation of cell surface receptors during the transmission of the TGF-β signaling pathway causes the intracellular Smad proteins to get phosphorylated and complex to form. After being translocated into the nucleus, the activated Smad complexes work with other nuclear cofactors to control target gene transcription [35]. TGF-β has the ability to regulate extracellular matrix turnover, which may alter the balance of collagen synthesis and degradation [36,37,38]. The gene expression of human scleral fibroblasts (HSFs) exposed to ultraviolet A (UVA) radiation was investigated by Yu-ting Hsiao et al. The production of type I collagen was decreased by the downregulation of TGF-β1 and Smad3, which were hypothesized to be possible upstream regulators [39]. Jobling et al. found that only one day after myopia developed, the TGF-β1, -β2, and -β3 genes were significantly downregulated in the tree shrews’ scleral tissue. Using primary scleral fibroblasts, in vitro research has shown that collagen synthesis decreases when the levels of the three TGF-βs reduce [40]. Two proteins were identified by this proteomics analysis: TGFB1 (FC = 1.02) and TGFB2 (FC = 1.18). At the same time, MMP2 protein expression was significantly downregulated (FC = 0.89), but COL1A1 protein expression remained unchanged (FC = 1.03). In animal models of myopia, two characteristics have been noted: increased MMP-2 levels and decreased collagen levels [41,42,43]. The current findings suggest that terahertz radiation can induce changes in the expression of extracellular matrix-related proteins and genes in scleral fibroblasts, potentially resulting in imbalances between collagen synthesis and degradation within these cells. This finding is consistent with the characteristics exhibited by scleral fibroblasts during myopia progression. Nevertheless, such cellular-level investigations are inadequate to clarify the causal relationship between terahertz radiation and scleral tissue effects. Additional validation is necessitated across a broader spectrum of terahertz radiation parameters, integrating scleral tissue-level evaluations such as measurements of scleral thickness, collagen turnover, extracellular matrix deposition, and biomechanical analyses.

## 5. Conclusions

This study examined the global expression of proteins within HFSFs exposed to 0.1 THz radiation. Proteomic analysis revealed that 0.1 THz radiation did not induce significant apoptosis or upregulated expression of heat shock proteins. Terahertz radiation upregulated the expression of three cytoskeleton-associated keratin proteins in HFSFs, which may enhance cellular morphological stability and damage resistance but increase the propensity for keratinizing diseases. Terahertz radiation downregulates the expression of multiple ribosomal and mitochondrial proteins in cells, which may reduce cellular protein synthesis and induce various neurodegenerative diseases. Terahertz radiation inhibits the activation of the TGF-β/Smad signaling pathway; however, prolonged radiation exposure reduces collagen synthesis and accelerates collagen degradation. These results indicate that the effects of 0.1 THz radiation on cellular functions are multifaceted, yet they may have the potential to influence myopia development by regulating intracellular protein expression and gene expression.

## Figures and Tables

**Figure 1 cells-14-01512-f001:**
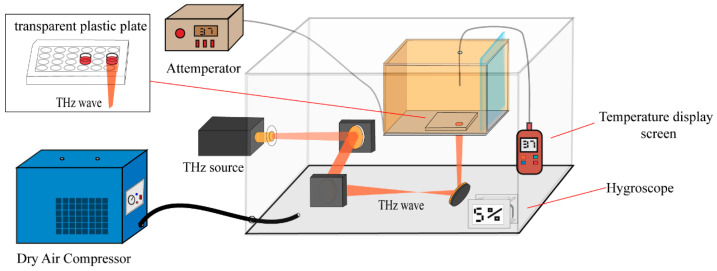
The schematic diagram of the terahertz radiation system.

**Figure 2 cells-14-01512-f002:**
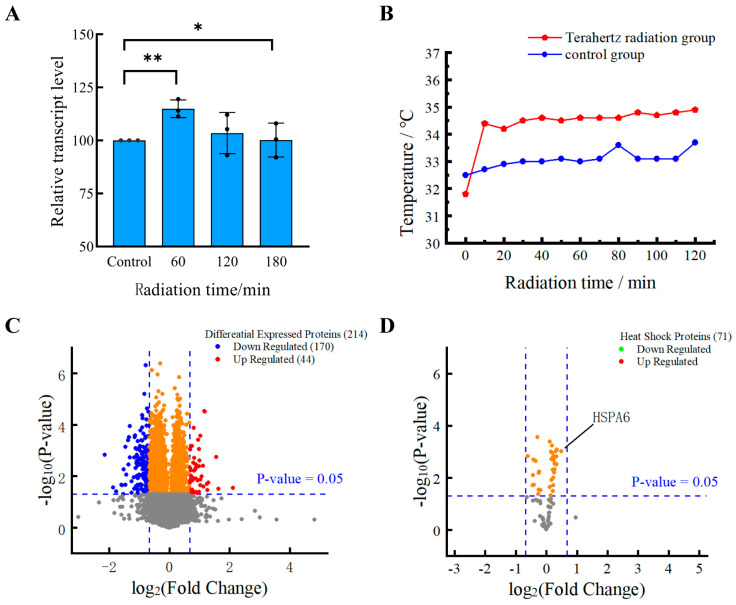
The effects of terahertz radiation on cellular activity and protein expression levels. (**A**) CCK-8 assay-measured percentage change in cell viability following 60, 120, and 180 min of terahertz radiation exposure; Data are expressed as mean ± standard deviation. Significance: * denotes *p* < 0.05, ** denotes *p* < 0.01. (**B**) Temperature variations in the culture medium over a 120 min exposure to 0.1 THz radiation. (**C**) The volcano graphic depicts global protein expression in HFSFs cells exposed to terahertz radiation. The rightmost area (red) displays 44 upregulated proteins, whereas the leftmost part (blue) displays 170 downregulated proteins. Proteins with *p* ≥ 0.05 (orange) and those with *p* ≥ 0.05 and FC ≥ 0.625 or FC ≤ 1.6 (grey) were excluded from the analysis. 1.6 up and down changes are represented by the vertical lines, and a *p*-value of 0.05 (*n* = 3 per group) is shown by the horizontal lines; (**D**) The expression of heat shock proteins (HSPs) in HFSF cells exposed to terahertz radiation is displayed in the volcano plot. The GeneCards database provided 109 heat shock protein data (correlation score > 18) that were compared to the proteins identified in this study.

**Figure 3 cells-14-01512-f003:**
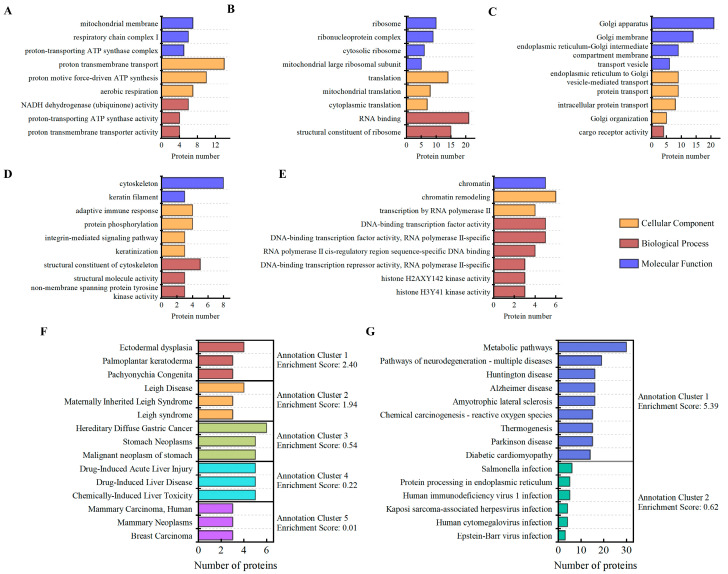
Functional enrichment analyses of differentially expressed proteins. (**A**–**E**) DEPs in the control and terahertz radiation groups were analyzed using Gene Ontology (GO) enrichment analysis. In the DAVID database, these DEPs were employed for functional annotation clustering. The number of enriched proteins was used to reorder biological processes, cellular components, and molecular functions after annotation clusters with the same “cellular component” were combined. (**A**) ribosomes, (**B**) mitochondria, (**C**) intermediates of the golgi apparatus, (**D**) cytoskeleton, and (**E**) chromatin; (**F**,**G**) Disease enrichment and Kyoto Encyclopedia of Genes and Genomes (KEGG) analyses for the DEPs were performed using the DAVID database (https://davidbioinformatics.nih.gov/, access Date: 2 June 2025); The enrichment scores for the top 5 disease annotation clusters are as follows: 2.04 (red), 1.94 (orange), 0.54 (green), 0.22 (cyan) and 0.01 (purple). The enrichment scores for the two signal pathway annotation clusters are: 5.39 (blue) and 0.62 (green). The y-axis shows the GO, KEGG and Disease functional categories, while the x-axis shows the number of DEPs corresponding to each function.

**Figure 4 cells-14-01512-f004:**
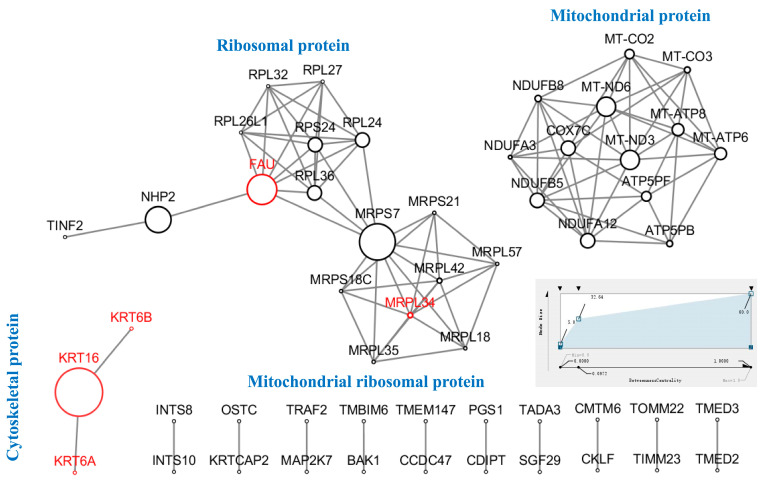
Protein–protein interaction (PPI) analysis for differentially expressed proteins. The signaling network among the DEPs was shown by PPI analysis. The DEPs were subjected to network analysis using STRING (www.string-db.org). Red indicates proteins that are up-regulated, while black indicates proteins that are down-regulated. The relationship between nodes was represented using the betweenness centrality measure, which also dictated the size.

**Figure 5 cells-14-01512-f005:**
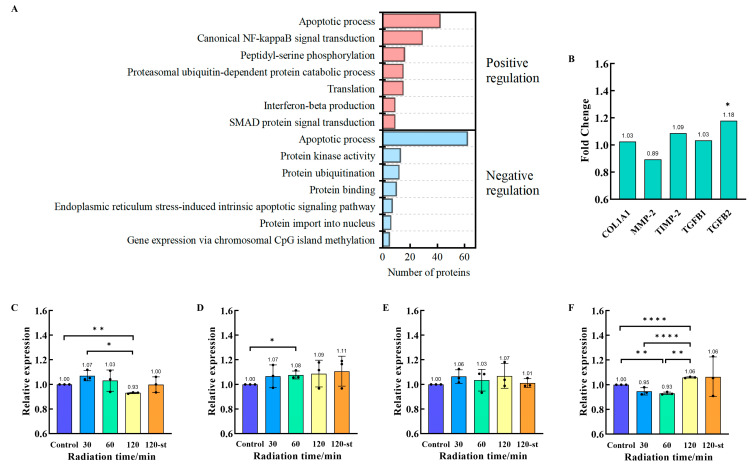
0.1 THz radiation inhibits collagen synthesis and degradation in HFSFs. (**A**) Gene Ontology (GO) enrichment analysis for DEPs (FC ≥ 1.2, FC ≤ 0.83, *p* < 0.05). The top 7 biological processes with positive and negative regulation terms. The top seven biological processes involved in positive (red) and negative (blue) regulation. (**B**) COL1A1, MMP-2, TIMP-2, TGFB1, and TGFB2 protein expression levels in HFSF cells as determined by proteomics analysis. (**C**–**F**) COL1A1, MMP-2, TIMP-2, and TGF-β2 gene expression levels in HFSF cells after 30 (blue), 60 (cyan), or 120 (yellow) min of terahertz radiation exposure. The 120-st group (orange) is made up of HFSFs that had been exposed to terahertz radiation for 120 min, then incubated for twelve hours before q-PCR analysis. Data are shown as mean the mean ± standard deviation (SD). Significance: * denotes *p* < 0.05, ** denotes *p* < 0.01, and **** denotes *p* < 0.0001. compared with control group.

**Table 1 cells-14-01512-t001:** qPCR primer sequences for COL1A1, MMP-2, TIMP-2, TGF -β2 and GAPDH from human.

Gene	Forward Primer (5′-3′)	Reverse Primer (5′-3′)
GAPDH	GCACCGTCAAGGCTGAGAAC	TGGTGAAGACGCCAGTGGA
COL1A1	CCCGGGTTTCAGAGACAACTTC	TCCACATGCTTTATTCCAGCAATC
MMP2	CTCATCGCAGATGCCTGGAA	TTCAGGTAATAGGCACCCTTGAAGA
TGFB2	TTACACTGTCCCTGCTGCACTT	GGTATATGTGGAGGTGCCATCAA
TIMP2	GGAGCACTGTGTTTATGCTGGAA	GAGACATGCGCAGTCTGCTTG

## Data Availability

Data reported in this paper will be shared by the lead contact upon request.

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
