# Peer review of "Using 0.1 THz Radiation Regulates Collagen Synthesis Through TGF-β/Smad Signaling Pathway in Human Fetal Scleral Fibroblasts"

_cells, 2025, doi:10.3390/cells14191512_

Round 1
Reviewer 1 Report
Comments and Suggestions for Authors
The work by Dr. Wang is devoted to studying the effects of 100 GHz radiation on cell cultures. The presented work is a classic omics study demonstrating changes in the expression of a number of genes and in protein content under the influence of radiation. The experimental design is well thought-out, the manuscript is easy to read, and it may be of interest to a wide range of readers, especially in light of the active integration of terahertz radiation in medicine, communications, and other industries. At the same time, minor revisions are required regarding the description of the statistical analysis.
Minor claims:
- In the "Materials and Methods" section, it is necessary to include a paragraph devoted to statistical data analysis.
- It is not entirely clear how many aliquots were used or how many replicates were carried out for the irradiation experiments. The authors must clearly indicate the sample size, which is mentioned only once in the manuscript (Line 186).
- The authors use the Student's t-test but do not report whether its assumptions were met, primarily the assumption of homoscedasticity (which could be checked using, for example, Levene's or Fisher's test). If this assumption is violated, it would be preferable to use the Welch test.
Reviewer 2 Report
Comments and Suggestions for Authors
In the proposed study, the authors investigate how 0.1 THz radiation affects human fetal scleral fibroblasts (HFSFs), with a particular focus on collagen synthesis, extracellular matrix remodeling, and the TGF-β/Smad signaling pathway. The main objective was to determine whether terahertz radiation contributes to the development of myopia by altering protein and gene expression patterns in these cells.
To address this, HFSFs were cultured under standard conditions and then exposed to 0.1 THz radiation at a power density of 44 mW/cm² for varying durations between 30 and 180 minutes. Cell viability was measured using the CCK-8 assay, while large-scale proteomic profiling was performed through LC-MS/MS. The differentially expressed proteins were analyzed using bioinformatics methods, including gene ontology and KEGG pathway enrichment as well as protein–protein interaction network analysis. In addition, quantitative PCR was used to validate changes in the expression of extracellular matrix–related genes such as COL1A1, MMP-2, TIMP-2, and TGF-β2.
The results showed that short-term exposure to terahertz radiation increased cell viability, peaking after 120 minutes, but this effect diminished with longer exposure. Proteomic analysis revealed 214 differentially expressed proteins out of 3055 screened, with nearly 80 percent being downregulated and about 20 percent upregulated. Downregulated proteins were mainly associated with ribosomes and mitochondria, suggesting a reduction in protein synthesis and mitochondrial respiration, while upregulated proteins included keratins such as KRT16, KRT6A, and KRT6B, which are linked to stress responses and tissue remodeling. Heat shock proteins did not show significant changes, indicating that the observed effects were not due to thermal stress but to non-thermal biological mechanisms. Pathway analysis indicated that terahertz radiation disrupted mitochondrial function and modulated the TGF-β/Smad signaling pathway. The gene expression analysis further confirmed time-dependent alterations in collagen-related pathways, with changes persisting up to 12 hours after radiation exposure.
According to the authors, the study demonstrated that 0.1 THz radiation alters protein and gene expression in human fetal scleral fibroblasts in a time-dependent manner. It enhances cell viability in the short term but eventually suppresses collagen synthesis and promotes its degradation through the TGF-β/Smad pathway. These findings suggest that terahertz radiation may contribute to myopia development by influencing scleral extracellular matrix remodeling and highlight the potential of non-thermal mechanisms in mediating its biological effects.
I have several concerns regarding the methods and the analysis of results.
Method section:
1) several core exposure-system details are missing or under-specified for reproducibility and dose interpretation: (i) exact source model (only “TeraSense… IMPATT diodes” is given, no model/serial), (ii) plate material transmission (24-well polystyrene will attenuate/reflect THz—no transmission or path-loss quantification is provided), (iii) beam profile and uniformity at the cell monolayer, (iv) dish-to-source distance and angle, (v) in-situ temperature of the culture medium during irradiation. The chamber is “37 °C” and humidity <5%, but there are no direct temperature traces in wells or medium; the claim of non-thermal conditions is inferred from HSP results, not dosimetry. These omissions limit interpretability of “non-thermal” conclusions and complicate replication.
2) Proteomics indicates n = 3 per group (volcano-plot caption) — acceptable for discovery but underpowered for small effects and multiple testing; explicit biological vs. technical replicate distinction is not always clear. CCK-8 says “each experimental sample was repeated three times,” but does not unambiguously state biological replicate count (wells vs. independent cultures). qPCR states “at least three biological replicates,” which is minimal. A priori power estimates are not reported.
3) The manuscript references Supplementary Tables S1–S4 and a placeholder MDPI link; in this file, those materials are not accessible. Because many conclusions (HSP list, clustering, PPI betweenness, DAVID clusters) depend on supplements, availability is essential for verification. Ensure that the supplements are complete and deposited.
Results section:
4) From 3,055 quantified proteins, 214 DEPs are called (FC≥1.6/≤0.625, p < 0.05) with ~79% down-regulated, interpreted as broad suppression of cellular processes. Without FDR-controlled differential testing, these lists are unreliable; volcano-plot thresholds do not substitute for multiple-testing correction. The conclusion “primarily affects processes via regulating protein levels” is plausible but needs more stringent stats and effect-size context.
5) GO/KEGG/PPI findings (mitochondria, ribosome, cytoskeleton; keratin upregulation; OXPHOS components down) are internally consistent, but disease-name enrichments (e.g., “Leigh syndrome,” “breast carcinoma”) are over-interpreted from annotation terms and should not be presented as potential THz-induced diseases without direct evidence. Keep these as hypothesis-generating signals only.
Discussion:
6) The discussion appropriately situates findings within prior THz literature but occasionally overreaches: disease-term enrichments are discussed as if indicative of potential disease induction; myopia implications are asserted (including contradictory lines that THz could prevent myopia at 60 min yet accelerate it by 120 min) without direct functional readouts (collagen protein turnover, ECM deposition, biomechanical assays). Please tone these to speculative/hypothesis-generating and clearly mark uncertainties.
Round 2
Reviewer 2 Report
Comments and Suggestions for Authors
The authors have responded adequately to all my comments.
The article is now acceptable for publication.